# DIFFERENTIABLE AVERAGE PRECISION LOSS IN DETR

## ABSTRACT

Average Precision (AP) and mean AP (mAP) remain the dominant measures for evaluating object detectors, yet most training objectives optimize surrogates that are not well aligned with these ranking-based measures. Most prior AP surrogates either rely on pairwise comparisons, incurring quadratic complexity, or only optimize classification, necessitating additional localization losses. We introduce differentiable average precision (DAP) loss, a smooth AP loss that directly optimizes a differentiable approximation to COCO-style mAP for one-to-one detectors. Our key idea is to (i) replace non-differentiable sorting by modeling detection scores as continuous distributions and sweeping a series of thresholds to obtain a differentiable precision–recall curve, and (ii) use interpolation techniques to optimize localization task. This yields a differentiable mAP approximation with linear time $(O(N))$ in the number of predictions, enabling seamless integration with Hungarian matching. We prove that, with respect to prediction scores, the gradient of DAP is sign-consistent—positive for positives and negative for negatives. Empirically, fine-tuning pretrained DETR-family models with DAP for a small number of epochs delivers consistent COCO mAP gains without auxiliary losses or architectural changes. DAP is simple to implement, computationally efficient, reduces hyperparameter tuning, and bridges the gap between training and evaluation for one-to-one detection. From-scratch training also delivers modest but positive improvements, albeit smaller than those obtained through fine-tuning.

## 1 INTRODUCTION

Object detection requires not only recognizing what is present in an image but also localizing it precisely. While modern detectors have made impressive progress, they are typically trained with a multi-task objective that decouples classification and localization into separate losses. This design simplifies optimization via gradient descent but introduces non-trivial hyperparameter tuning to balance loss weights, and it often leaves a gap between what is optimized during training and what is evaluated at test time. Average Precision (AP) and mean Average Precision (mAP) remain the de facto evaluation measures for detection. They jointly reflect classification quality and localization quality through precision–recall computation across intersection-over-union (IoU) thresholds. However, AP/mAP is intrinsically non-differentiable due to its reliance on sorting, thresholding, and discrete matching, making it difficult to optimize directly. Recent ranking-based surrogates attempt to close this gap by optimizing score ordering between positives and negatives (Chen et al., 2019; Oksuz et al., 2021; Yavuz et al., 2024b; Brown et al., 2020; Rolínek et al., 2020). Yet most such methods either (i) incur $O(N^2)$ complexity due to pairwise comparisons, which limits computation to per-image ranking losses and is hard to integrate into Hungarian bipartite matching, or (ii) primarily target classification, requiring additional localization losses and extra hyperparameters. Efforts to jointly consider localization and classification via aLRP (Oksuz et al., 2018; 2020) optimize a different objective and thus do not directly target AP. The Parameterized AP Loss (Chenxin et al., 2021) introduces auxiliary components such as reinforcement learning, increasing system complexity. Consequently, an efficient, practical objective that directly optimizes a close, differentiable proxy to COCO-style mAP remains elusive.

To this end, we introduce DAP, a smooth AP loss that directly optimizes a differentiable approximation to COCO-style mAP for one-to-one detectors such as the DETR family. It models scores as continuous densities and sweeps soft thresholds to form a differentiable precision–recall curve,

eliminating pairwise comparisons and enabling linear-time integration with Hungarian matching. Localization is incorporated via IoU interpolation, coupling classification and localization within a single objective and simplifying tuning.

DAP exhibits desirable optimization behavior. With respect to prediction scores, its gradients are sign-consistent—positive for positives and negative for negatives—whereas many ranking-based surrogates may exhibit sign inconsistencies under certain configurations. To guarantee sign-consistent gradients, we impose three mild score distributional principles: (i) non-negativity, (ii) unit mass (normalization), and (iii) tail monotonicity that preserves the score ordering (higher scores have larger tail probabilities at every threshold). Under these conditions, we prove sign-consistent gradients (see Section 3 and Appendix A.2).

When fine-tuning pretrained DETR-style models for a small number of epochs, DAP yields consistent COCO-style mAP gains without architectural changes or auxiliary losses.

The main contributions of this paper are summarized as follows:

1. We introduce DAP, a smooth loss that directly approximates COCO-style mAP for one-to-one detectors, with a linear-time formulation that eliminates pairwise comparisons and integrates naturally with Hungarian matching;

2. We prove sign-consistent gradients with respect to scores and jointly optimize classification and localization without auxiliary losses, with simple sufficiency conditions on the score density and a practical Gaussian instantiation;

3. We demonstrate consistent fine-tuning gains across multiple DETR-style detectors, incurring only minor training overhead and no additional inference cost.

## 2 RELATED WORK

### 2.1 DETR FOR OBJECT DETECTION

The pioneering work DETR (Carion et al., 2020) employed set-based prediction to achieve end-to-end object detection. Its simplicity and outstanding performance have led to numerous proposed extensions. Deformable DETR (Zhu et al., 2020) introduced a multi-scale deformable self/cross-attention mechanism that selectively focuses on a small number of key sampling points in the reference boxes. Compared to DETR, Deformable DETR significantly accelerates convergence and achieves improved performance. DAB-DETR (Liu et al., 2022) and DN-DETR (Li et al., 2022) demonstrated that the query formulation in the decoder can significantly impact DETR's performance. DINO-DETR (Zhang et al., 2023) achieved shorter training times and better performance, by addressing the instability issues of the one-to-one matching problem. RT-DETR (Zhao et al., 2024), on the other hand, has extended DETR into the realm of real-time object detection, enabling broader practical applications. Building on these, this paper integrates DAP-loss with DETR and its variants, further enhancing their detection capabilities. Co-DETR (Zong et al., 2023) simultaneously employs both one-to-one and one-to-many matching strategies in its training, which not only improves performance and convergence speed but also preserves the end-to-end detection capability.

### 2.2 AP AS A LOSS FOR OBJECT DETECTION

Average Precision (AP), which takes into account both classification and localization tasks, is the most commonly used evaluation measure in object detection. However, due to its non-differentiability and non-convexity, AP cannot be directly used as an optimization objective in object detection. Several methods have been proposed to tackle the challenge of optimizing AP loss in object detection. AP-loss (Chen et al., 2019) and its extensions (Pu et al., 2024; Xu et al., 2022; Yavuz et al., 2024b) utilize error-driven updates, employing Rankloss as a classification loss to indirectly optimize AP. While effective, these methods often decouple the optimization of the metric from the direct supervision of localization regression. Early works like Song et al. (2016) formulate AP optimization as a structured prediction problem using a loss-augmented objective. While Henderson & Ferrari (2017) propose an approach to optimize AP that explicitly incorporates the Non-Maximum Suppression (NMS) procedure, their method is limited to the ranking and selection of fixed candidate windows. Specifically, although their loss function accounts for the suppression of overlapping

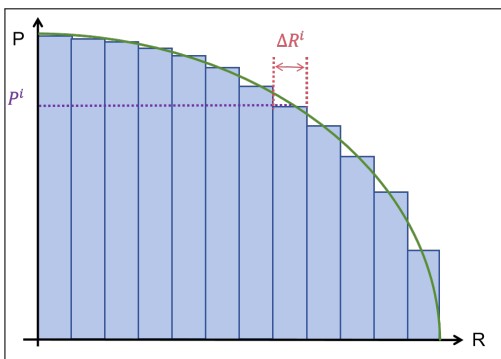

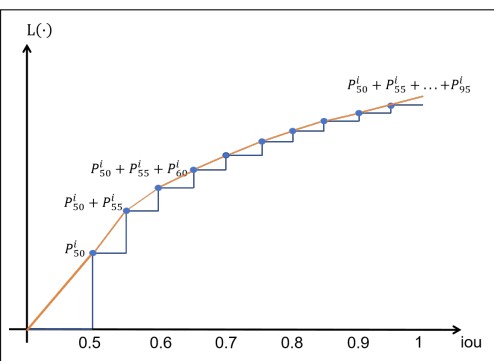

Figure 1: **Rectangle Integration for AP Approximation.** This figure demonstrates the approximation of the area under the PR curve using $N$ rectangles.

Figure 2: Curves of $L(\cdot)$ (blue) and its smoothed version $L_{int}(\cdot)$ (orange)

boxes (a localization aspect), they do not perform bounding-box regression to refine the coordinates of the proposals. Alternatively, Mohapatra et al. (2018) leverages reinforcement learning to improve the original classification loss starting from AP. Although these methods have achieved some valuable results in optimizing AP loss, they still have limitations. A key issue is that they primarily replace the classification loss with one based on the AP metric, without adequately addressing the localization task. The Parameterized AP Loss (Chenxin et al., 2021) advances the approach by utilizing reinforcement learning to parameterize the non-differentiable aspects of AP, allowing for the joint optimization of localization and classification tasks. However, it necessitates a computationally expensive parameter search phase prior to detector training, which increases the overall cost and hinders efficient fine-tuning.

## 3 METHOD

We aim to directly optimize AP in DETR-style detectors—end-to-end, one-to-one architectures based on bipartite matching.

### 3.1 PRELIMINARY

**DETR:** Detection Transformers (DETR) (Carion et al., 2020) passes the input image $I$ through a backbone network and a transformer encoder to obtain a series of enhanced feature embeddings $X$. These feature embeddings are then combined with a set of $N_q$ object query embeddings $Q$ and passed through a transformer decoder to produce $N_q$ output predictions. Finally, DETR performs a one-to-one bipartite matching between the predictions and the ground-truth annotations $GT$ for bounding boxes and labels, associating each ground truth annotation with a prediction instance with the minimum matching cost. Prediction instances matched to ground truth annotations are classified as **positive** samples, while unmatched instances are considered **negative**. The corresponding losses are then computed separately for the positive and negative samples. Our method adopts the same matching to determine positives and negatives during training. DETR employs a one-to-one matcher, where each ground-truth instance is matched with only one positive prediction. This is a key factor that enables DETR to perform end-to-end inference.

### 3.2 AP MEASURE

The AP in object detection serves as a critical benchmark for evaluating the performance of a detector, derived from the area under the Precision-Recall (PR) curve (generated by varying the score threshold). The final AP is computed by averaging these areas over uniformly sampled IoU thresholds ranging from 0.50 to 0.95 with a step size of 0.05.

As shown in Fig. 1, we approximate the definite integral of the PR curve with the rectangular method.

$$AP_\alpha = \int_0^1 P_\alpha(x) \, dR_\alpha(x) = \int_1^0 P_\alpha(x) \frac{dR_\alpha(x)}{dx} dx \approx \frac{1}{G} \sum_{i=0}^{N-1} P_\alpha^i \Delta T_\alpha^i, \tag{1}$$

where $\alpha$ represents the IoU threshold, and $x \in [0,1]$ denotes the score threshold. The value $G$ indicates the total count of ground truth. $T^i$ and $P^i$ represent the number of true positives and the precision at the score threshold $x = \frac{i}{N}$, respectively. The term $\Delta T_\alpha^i$ denotes the change in the number of positive samples (definitions: $T^i - T^{i+1}$). Notably, since Recall is defined as $\frac{T}{G}$, the differential $dR_\alpha(x)$ in the continuous domain relates to the discrete term $\frac{\Delta T_\alpha^i}{G}$, which introduces the normalization factor $\frac{1}{G}$ in the summation.

With this, AP can be calculated as follows:

$$AP = \frac{1}{10}(AP_{50} + AP_{55} + \ldots + AP_{90} + AP_{95}) \approx \frac{1}{10G} \sum_{i=0}^{N-1} \sum_{\alpha \in \mathcal{S}_\alpha} P_\alpha^i \Delta T_\alpha^i, \tag{2}$$

where $\mathcal{S}_\alpha$ is the set of IoU thresholds, defined as $\{0.5, 0.55, \ldots, 0.90, 0.95\}$. $T_\alpha^i$ denotes the number of positive samples. By leveraging the sampling property of the impulse function, we rewrite Equation 2 in the form of a summation over positive samples, obtaining:

$$AP \approx \frac{1}{10G} \sum_{i=0}^{N-1} \sum_{b \in \mathcal{T}} L(b_{iou}, i) \cdot \chi^{(N)}(b_{score}, i), \tag{3}$$

where $\mathcal{T}$ is the set of positive outputs, *i.e.,* those matched with ground truth annotations; $b_{score}$ and $b_{iou}$ represent the score of the predicted bounding box $b$ and its IoU with the ground-truth, respectively. The score-bin indicator function $\chi(\cdot)$ and the localization score function $L(\cdot)$ are defined as,

$$L(b_{iou}, i) = \sum_{\alpha \in \mathcal{S}_\alpha} \mathbb{I}(b_{iou} > \alpha) \, P_\alpha^i, \tag{4}$$

$$\chi^{(N)}(b_{score}, i) = \int_{\frac{i}{N}}^{\frac{i+1}{N}} \delta(b_{score} - t) \, dt = \mathbb{I}(\frac{i}{N} \le b_{score} < \frac{i+1}{N}), \tag{5}$$

where $\mathbb{I}(\cdot)$ is the indicator function, and $\delta(\cdot)$ denotes the Dirac delta distribution (impulse). Under a score threshold of $\frac{i}{N}$, the visualization of the function $L(\cdot)$ is depicted by the blue line in Figure 2.

As shown in Equation 3, the AP is determined by the localization score function $L(\cdot)$ and function $\chi(\cdot)$. In following sections, we will discuss $L(\cdot)$ and $\chi(\cdot)$ from the perspectives of localization and classification.

### 3.3 LOCALIZATION

From Equation 3, it follows that since $\chi(\cdot)$ is independent of IoU, the optimization signal for localization must come entirely from $L(\cdot)$. Therefore, in this subsection, we focus solely on the function $L(\cdot)$.

Consider a predicted bounding box matched with a ground truth, which serves as a positive sample during training. According to Equation 4, $L(\cdot)$ is determined by 10 points: $(0.5, P_{50}^i), (0.55, P_{50}^i + P_{55}^i), \ldots, (0.95, P_{50}^i + P_{55}^i + \ldots + P_{95}^i)$. It is clear that within the range of $IoU \in [0,1]$, the gradient of function $L(\cdot)$ with respect to IoU either does not exist or is zero. Consequently, the task of optimizing localization cannot be effectively addressed using the backpropagation algorithm. While methods like Sigmoid approximation, Error-Driven learning, and Finite Difference are commonly used to handle step functions, some smooth surrogates (e.g., Sigmoid, Tanh) suffer from vanishing gradients. We instead adopt linear interpolation, leveraging our access to the step function's endpoints. This choice offers a simple yet effective solution with superior gradient characteristics, avoiding the saturation issues of some non-linear alternatives. Specifically, as shown by the red line

in Figure 2, we construct a piecewise-linear interpolation $L_{int}(\cdot)$, adding the origin and extending the tail to 1.0, which supplies non-zero gradients across $IoU \in [0, 1]$. The interpolation function is defined as follows:

$$L_{int}(b_{iou}, i) = y_k^i + \frac{y_{k+1}^i - y_k^i}{x_{k+1} - x_k}(b_{iou} - x_k), \quad \text{for } b_{iou} \in [x_k, x_{k+1}), \tag{6}$$

where $b_{iou}$ denotes the IoU. The set of anchor points $\{(x_k, y_k^i)\}$ consists of the origin ($x_0 = 0, y_0^i = 0$), the 10 cumulative points listed above (where $x_k \in \{0.5, 0.55, \ldots, 0.95\}$), and the endpoint extension at $x_{11} = 1.0$ (where $y_{11}^i = 2y_{10}^i - y_9^i$).

## 3.4 CLASSIFICATION

The precision terms in $L(\cdot)$ and the hard binning in $\chi(\cdot)$ yield zero gradients almost everywhere, hindering training.

To derive an appropriate gradient, we model the scores of each predicted bounding box using a continuous probability distribution.

Given a score threshold $x$, the probability that a predicted bounding box $b$ is classified as positive is represented by the **complementary cumulative distribution function** (tail distribution), defined as follows:

$$\Pr(b, x) = \int_x^1 f^b(y) dy, \tag{7}$$

where $y$ represents the score variable, and $f^b(\cdot)$ denotes the Probability Density Function (PDF) for $b$. When $f^b(\cdot)$ is represented as an impulse function $\delta(\cdot)$, the equation above becomes equivalent to the original model, as follows:

$$\Pr(b, x) = \int_x^1 \delta(y - b_{score}) \, dy = \mathbb{I}(b_{score} \geq x), \tag{8}$$

where $\mathbb{I}(\cdot)$ is the indicator function. This result corresponds exactly to the standard hard-thresholding mechanism (outputting 1 if score $b_{score} \geq x$, else 0).

To establish design guidelines for $f^b(\cdot)$, we calculated the partial derivatives of the $AP$ with respect to the scores of positive and negative samples, yielding the following equations. The detailed derivation can be found in Supplementary Material A.1:

$$\frac{\partial AP_\alpha}{\partial s_t} = \frac{1}{G} \int_0^1 \frac{\partial f^t(x)}{\partial s_t} \cdot P(x) + \frac{\frac{\partial \Pr(t,x)}{\partial s_t} \cdot \text{FP}(x)}{(\text{T}(x) + \text{FP}(x))^2} \cdot \sum_{b \in \mathcal{T}} f^b(x) dx, \tag{9}$$

$$\frac{\partial AP_\alpha}{\partial s_n} = \frac{1}{G} \int_0^1 \frac{-\frac{\partial \Pr(n,x)}{\partial s_n} \cdot \text{T}(x)}{(\text{T}(x) + \text{FP}(x))^2} \cdot \sum_{b \in \mathcal{T}} f^b(x) dx, \tag{10}$$

where $\text{FP}(x)$ and $\text{T}(x)$ respectively represent the false positive and true positive at a score threshold of $x$ and an IoU threshold of $\alpha$. Additionally, $s_t$ and $s_n$ denote the scores of a positive sample and a negative sample, respectively. And $\mathcal{T}$ is the set of all positive samples.

In the original model (*i.e.*, $f^b(x)$ is impulse function), both $\frac{\partial \Pr(t,x)}{\partial s_t}$ and $\frac{\partial \Pr(n,x)}{\partial s_n}$ are zero except at a finite number of points. This characteristic significantly affects the gradients $\frac{\partial AP_\alpha}{\partial s_t}$ and $\frac{\partial AP_\alpha}{\partial s_n}$, causing the derivatives of the classification scores to become zero.

To facilitate training with the gradient descent algorithm, we aim for the function $f^b(\cdot)$ to satisfy the following characteristics: *i.* $\frac{\partial AP_\alpha}{\partial s_t} > 0, \frac{\partial AP_\alpha}{\partial s_n} < 0$ (Sign Consistency); *ii.* $f^b(\cdot)$ is an approximation of $\delta(\cdot)$.

For the first characteristic, a simple and effective approach is to ensure that each term in Equation 9 is greater than or equal to 0, while each term in Equation 10 is less than or equal to 0. We obtain the following specific and simple conditions that enable $f^b(\cdot)$ satisfy this characteristic. The relevant mathematical proof can be found in Supplementary Material A.2.

The three conditions for the distribution $f^b(\cdot)$ are as follows:

1. $f^b(x) \geq 0$ for all $x \in [0, 1]$;
   *Condition 1* ensures that the distribution is non-negative across the entire range.

2. $\int_0^1 f^b(x)\,dx = 1$;
   *Condition 2* ensures that the distribution is properly normalized, such that the total probability sums to 1.

3. $\int_a^1 f^{b_1}(x)\,dx > \int_a^1 f^{b_2}(x)\,dx,$ for $s_1 > s_2$ and for all $a \in (0, 1)$.
   *Condition 3* ensures that a bounding box with a higher score $s_1$ will have a higher probability of being a true positive compared to a box with a lower score $s_2$, thereby preserving the ranking of boxes based on their scores.

For the second characteristic, which states that $f^b(\cdot)$ is an approximation of the impulse function, we use a normalized Gaussian function to approximate the impulse function, setting the mean to the classification score $s$ and using a constant standard deviation $\sigma$. Clearly, probability distribution $\mathcal{N}(x; b_{score}, \sigma)$ satisfies all the aforementioned requirements of $f^b(\cdot)$.

Thus, in our proposed method, we use $f^b(\cdot)$ to replace $\delta(\cdot)$ from Equation 3. To make $\chi(\cdot)$ differentiable, we replace the Dirac delta function $\delta(\cdot)$ with our chosen probability distribution $f^b(\cdot)$. This yields a new 'soft' binning function, $F(\cdot)$, defined as:

$$
F(\frac{i}{N}, b_{score}) = \begin{cases} \int_{-\infty}^{\frac{1}{N}} \mathcal{N}(x; b_{score}, \sigma)dx & \text{if } i = 0, \\ \int_{\frac{i}{N}}^{\frac{i+1}{N}} \mathcal{N}(x; b_{score}, \sigma)dx & \text{if } 0 < i < N - 1, \\ \int_{\frac{i}{N}}^{+\infty} \mathcal{N}(x; b_{score}, \sigma)dx & \text{if } i = N - 1. \end{cases} \tag{11}
$$

The visualization of bin mass $F(\cdot)$ and $\chi(\cdot)$, along with their comparison, is shown in Figure 5.

In summary, the differentiable approximation of Equation 3, referred to as DAP-loss in this paper, can be expressed as follows:

$$
\text{DAP-loss} = -1 \times DAP = \frac{-1}{10G} \sum_{i=0}^{N-1} \sum_{b \in \mathcal{T}} L_{int}(b_{iou}, i) \cdot F(\frac{i}{N}, b_{score}), \tag{12}
$$

where $L_{int}(\cdot)$ represents a differentiable localization score function, as elaborated in Subsection 3.3 and Figure 2.

**Contrast with pairwise ranking surrogates.** Many pairwise rank-based methods can violate sign consistency. For example, consider a toy list with a positive sample $t$ having score $s_t = 0.2$, another positive with score $\{0.1\}$, and a negative with score $\{0.99\}$. Under *SmoothAP* (Brown et al., 2020) with temperature $\tau = 0.01$, we empirically observe $\frac{\partial\,\text{SmoothAP}}{\partial s_t} < 0$, i.e., the objective pushes down the score of a true positive (verified numerically). It is important to clarify that SmoothAP is designed as a differentiable approximation of AP, which is an objective to be **maximized**. Therefore, a valid approximation should yield a positive gradient ($\frac{\partial\,\text{SmoothAP}}{\partial s_t} > 0$) to increase the score of a true positive. The observed negative gradient implies that the optimization objective erroneously drives the score of the true positive $t$ downwards. Similar violations can occur in other pairwise rank-based methods under skewed score configurations.

### 3.5 AP-Cost Matcher

In general, to make the matching results more suitable for model training, DETR and its variants use the same function to define the Hungarian matching cost function $L_{match}$ as that used for the model training loss. Consistent with previous approaches, we will introduce a Hungarian matching cost function to fit the DAP-loss.

In end-to-end object detection models, during training, the positive and negative samples are determined by the output of the matcher. This means that when computing the matching cost $L_{match}$, the

information regarding which samples are positive is unavailable, making it impossible to compute $P_\alpha^i$ in Equation 3.

To address this issue and compute $MP_\alpha^i$, the estimated precision used by the matcher, we employ a momentum-based averaging method to estimate $P$, which we refer to as the **momentum** mode. We update the momentum parameter at step $t$ using the following formula:

$$MP_\alpha^i(t+1) = m \cdot MP_\alpha^i(t) + (1-m) \cdot P_\alpha^i. \qquad (13)$$

In well-trained models, when the score threshold is sufficiently high (e.g., greater than 0.5), $P_\alpha^i$ approaches 1. This observation leads to a simplified formulation, setting $MP_\alpha^i = 1$, which is termed the **constant** mode.

We substitute $MP_\alpha^i$ for $P_\alpha^i$ in function $L_{int}$ of Equation 12 to obtain a cost function that is suitable for the Hungarian algorithm. Finally, for a predicted bounding box $b$ and a ground truth $gt$, the cost function for the bipartite matching can be expressed as,

$$L_{match}(b, g) = \sum_{i=0}^{N-1} \mathrm{ML}_{int}(IoU(b, gt), i) \cdot F(\frac{i}{N}, b_{score}), \qquad (14)$$

where $\mathrm{ML}_{int}(\cdot)$ is obtained by replacing $P_\alpha^i$ with $\mathrm{MP}_\alpha^i$ in $L_{int}(\cdot)$. Visualizations of $\mathrm{ML}_{int}(\cdot)$ under the Momentum mode and Constant mode can be found in Supplementary Material A.3.

### 3.6 TRAINING DETAILS

**Minibatch Training:** Minibatching is standard in deep learning and improves stability and convergence versus batch size 1. AP estimation is sensitive to batch size; very small batches can misestimate AP when per-image score ranges differ (e.g., the lowest score in $I_1$ exceeds the highest in $I_2$). Aggregating scores within a minibatch mitigates this effect. We report the impact of batch size in the next section.

**Interpolated AP:** Interpolated Average Precision (Interpolated AP) is widely used in object detection benchmarks, such as PASCAL VOC (Everingham et al., 2015) and MS COCO (Lin et al., 2014). Compared to standard AP, Interpolated AP is less sensitive to minor fluctuations in predicted scores and better aligns with practical needs, which is why it is more commonly employed today. For these reasons, we adopt Interpolated AP instead of the original version. This implies that precision increases as the score threshold increases, *i.e.,* $P(i) \leq P(j)$ if $i < j$.

## 4 EXPERIMENT

**Dataset:** We train on COCO 2017 (Lin et al., 2014) train (118K images) and report results on the val set (5K images). Unless noted, measures are COCO-style bbox mAP, averaged over IoU thresholds 0.50–0.95 (step 0.05).

### 4.1 MAIN RESULTS

#### 4.1.1 IMPROVEMENTS ON PRE-TRAINED BASELINES

**Experiments Settings.** We fine-tuned multiple backbones (He et al., 2016; Dosovitskiy, 2020; Liu et al., 2021) of several state-of-the-art (SOTA) methods (Carion et al., 2020; Zhao et al., 2024; Zhu et al., 2020; Chen et al., 2024; Pu et al., 2023; Yavuz et al., 2024a) using the DAP-loss. Unless otherwise specified, all experiments, including baseline follow the settings outlined below.

The learning rate for all experiments was set to one-tenth of that used in the baseline method and further reduced to one-hundredth at the last epoch. Furthermore, the training duration for fine-tuning is set to approximately 10% of the pre-training length, with a maximum cap of 12 epochs.

For all ori-loss experiments, the loss settings are consistent with the official open-source code of the corresponding methods. In the DAP-loss experiments, the detection loss (including both classification and localization) of the corresponding methods is replaced with the proposed DAP-loss, while auxiliary losses, such as the denoising loss, remain unchanged.

The standard deviation ($\sigma$) of the distribution $\mathcal{N}$ in the DAP-loss is uniformly set to 0.05 across all experiments, and the variable **N** in Equation 3 is consistently assigned a value of 256.

In this experiment, results before fine-tuning are denoted as $r_o$, and those fine-tuned with the original loss are $r_r$, the reported original loss result is $\max(r_o, r_r)$. Since these are trained models, $r_r$ is typically slightly lower than or on par with $r_o$.

Table 1: Detection results on COCO 2017 val under brief fine-tuning of pretrained models. "Ori Loss" denotes each method's original training loss; "DAP Loss" replaces it during fine-tuning. All other settings remain unchanged.

| Method | Backbone | epochs | DAP Loss | Ori Loss | AP | $AP_{50}$ | $AP_{75}$ |
|---|---|---|---|---|---|---|---|
| DETR | R50 | 12 | | ✓ | 42.2 | 62.5 | 44.6 |
| DETR | R50 | 12 | ✓ | ✓ | 42.7 | 63.0 | 45.1 |
| DETR | R50 | 12 | ✓ | | 43.4(+1.2) | 63.0 | 45.8 |
| DETR | R101 | 12 | | ✓ | 43.6 | 64.1 | 46.1 |
| DETR | R101 | 12 | ✓ | | 44.9(+1.3) | 64.3 | 48.0 |
| Deformable DETR | R50 | 8 | | ✓ | 46.2 | 64.9 | 50.2 |
| Deformable DETR | R50 | 8 | ✓ | | 47.3(+1.1) | 65.1 | 51.3 |
| RT-DETR | R50 | 8 | | ✓ | 53.1 | 71.4 | 57.5 |
| RT-DETR | R50 | 8 | ✓ | | 53.4(+0.3) | 71.6 | 57.6 |
| LW-DETR-large | ViT | 8 | | ✓ | 56.1 | 74.5 | 61.0 |
| LW-DETR-large | ViT | 8 | ✓ | | 56.3(+0.2) | 74.2 | 61.3 |
| LW-DETR-xlarge | ViT | 8 | | ✓ | 58.3 | 77.2 | 63.2 |
| LW-DETR-xlarge | ViT | 8 | ✓ | | 58.7(+0.4) | 76.6 | 63.7 |
| Co-DETR | R50 | 4 | | ✓ | 49.9 | 68.1 | 55.0 |
| Co-DETR | R50 | 4 | ✓ | | 50.8(+0.9) | 68.0 | 55.4 |
| Co-DETR | Swin-B | 4 | | ✓ | 57.5 | 76.0 | 63.4 |
| Co-DETR | Swin-B | 4 | ✓ | | 58.1(+0.6) | 76.2 | 63.4 |
| Co-DETR | Swin-L | 4 | | ✓ | 58.5 | 77.1 | 64.5 |
| Co-DETR | Swin-L | 4 | ✓ | | 59.0(+0.5) | 77.3 | 64.4 |
| Rank-DETR | Swin-T | 2 | | ✓ | 54.7 | 72.5 | 60.0 |
| Rank-DETR | Swin-T | 2 | ✓ | | 55.1(+0.4) | 72.7 | 60.0 |
| Rank-DETR | Swin-L | 2 | | ✓ | 58.2 | 76.7 | 63.9 |
| Rank-DETR | Swin-L | 2 | ✓ | | 58.4(+0.2) | 76.8 | 63.7 |
| BRS-Co-DETR | R50 | 2 | | ✓ | 50.1 | 67.4 | 54.6 |
| BRS-Co-DETR | R50 | 2 | ✓ | | 50.8(+0.7) | 67.8 | 55.4 |
| BRS-Co-DETR | Swin-L | 2 | | ✓ | 57.2 | 70.5 | 62.5 |
| BRS-Co-DETR | Swin-L | 2 | ✓ | | 57.6(+0.4) | 76.0 | 62.4 |

**Results and analysis.** Table 1 summarizes comparisons across DETR-family baselines. Fine-tuning with DAP yields consistent gains with minimal changes to training pipelines.

Experimental evaluations show that fine-tuning with our proposed DAP-loss, even with minimal adjustments, enhances the performance of end-to-end object detection models. Notably, DAP exhibits a certain performance advantage even when compared to other AP-based methods, such as Rank-DETR (Pu et al., 2023) and BRS-Co-DETR (Yavuz et al., 2024a). This advantage likely arises because, unlike error-driven methods, DAP supplies gradients that directly target coco-style AP optimization, resulting in more effective model training.

On DETR-R50, combining DAP with the original loss improves AP by 0.5 vs. using the original loss alone, yet remains 0.7 below DAP-only fine-tuning, suggesting that DAP's balance of classification and localization is most effective when used as the sole detection objective.

Table 2: Results of different matching functions.

| Backbone | Cost Function Mode | AP |
|---|---|---|
| R50 | original | 42.8 |
| R50 | momentum | 43.4 |
| R50 | constant | 43.4 |
| R101 | original | 44.6 |
| R101 | constant | 44.9 |

Table 3: Results of training from scratch. '†' denotes DAP-only in the final epoch.

| Method | Loss config | AP |
|---|---|---|
| Co-DETR | Baseline | 49.3 |
| Co-DETR | DAP only | 47.4 |
| Co-DETR | DAP + Baseline | 49.5 |
| Co-DETR | DAP + Box regression | 49.4 |
| Co-DETR† | DAP + Box regression | 49.8 |

## 4.2 HYPERPARAMETER AND COMPONENT ANALYSIS

In this subsection, we conduct a systematic analysis to assess the impact of different components and experimental settings of our method. Unless otherwise specified, all experimental configurations in this subsection adhere to those used in the DETR experiments described earlier, with the backbone network fixed as ResNet-50.

### 4.2.1 ANALYSIS ON PARAMETER $N$

In Equations 1 and 3, the parameter $N$ denotes the number of rectangles used to segment the area beneath the Precision-Recall (PR) curve. It is evident that as $N$ increases, the method proposed in this paper provides a closer approximation to AP.

Figure 3a illustrates how the COCO AP varies with $N$. The trend indicates a general increase in AP as $N$ rises, with stabilization occurring beyond a value of 32. This behavior aligns with our earlier discussion, indicating that a higher parameter $N$ leads to a more accurate approximation of the true AP for a set of images.

### 4.2.2 ANALYSIS ON STANDARD DEVIATION $\sigma$

We examined the standard deviation $\sigma$ of $\mathcal{N}$, the sole hyperparameter in DAP-loss that requires fine-tuning. A very large $\sigma$ can cause the function to deviate from the properties of the step function $\delta(\cdot)$ in Equation 3, while a very small $\sigma$ can result in too small gradients, potentially hindering training. Figure 3b presents COCO AP results for various $\sigma$ values, indicating that the best performance is achieved at $\sigma = 0.05$, with satisfactory results within the range of $[0.01, 0.1]$.

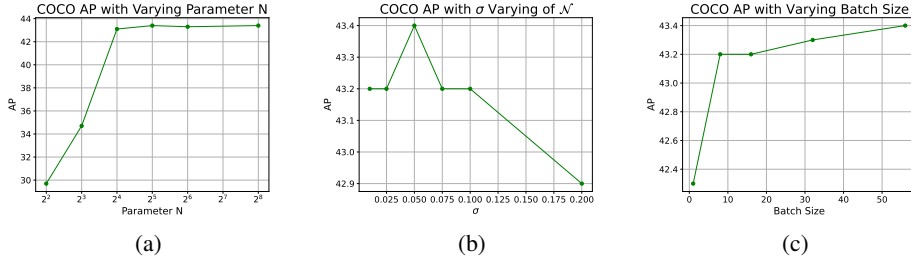

(a)         (b)         (c)

Figure 3: COCO 2017 validation results based on DETR.

### 4.2.3 ANALYSIS ON BATCH SIZE

As depicted in Figure 3c, optimal performance generally increases with batch size within the range of $[1, 56]$. Notably, at a batch size of 8, the performance is merely 0.2% AP below the optimal result. This trend is intuitive, as a very small batch size can lead to a significant discrepancy between the AP of the batch and that of the entire dataset.

#### 4.2.4 ANALYSIS ON THE HUNGARIAN MATCHER COST FUNCTION

In Subsection 3.5, we introduced two matching cost functions designed for DAP-loss. Table 2 reports the results obtained with these different matching functions. Both the Constant and Momentum modes show similar improvements and outperform the original mode. Considering the greater complexity of the Momentum mode, we recommend using the Constant mode.

#### 4.2.5 RESULTS OF TRAINING FROM SCRATCH

Table 3 shows that DAP-only training from scratch is suboptimal. We speculate that early low precision flattens the precision–recall surface and, as visualized in Figure 2, yields weak localization (regression) gradients that hinder convergence. Adding a box regression loss as an auxiliary objective improves AP but still trails DAP-only fine-tuning. See Appendix A.4 for more analysis.

## 5 SCOPE AND LIMITATIONS

Because COCO-style mAP matches at most one prediction to each ground-truth instance, DAP is designed for one-to-one detectors with bipartite matching (e.g., DETR) and is not directly applicable to one-to-many frameworks that allow multiple positives per ground-truth. Extending DAP would require a differentiable de-duplication or matching mechanism.

From-scratch training with DAP is less effective: early predictions yield weak gradients and slow convergence. Auxiliary losses can help initially, but joint training introduces additional hyperparameters and tuning overhead and, in our experiments, still underperforms DAP-only fine-tuning. In practice, we find that brief fine-tuning of pretrained one-to-one detectors with DAP offers better accuracy at lower cost.

## 6 CONCLUSION

We decompose AP's non-differentiability into localization and classification components, and construct a differentiable surrogate—Differentiable Average Precision (DAP)—using piecewise-linear interpolation over IoU and Gaussian score smoothing. We further define a Hungarian matching cost compatible with DAP for end-to-end detectors. DAP jointly optimizes localization and classification with minimal tuning (a single standard deviation parameter $\sigma$), and we prove sign-consistent score gradients with supporting derivations. Empirically, DAP fine-tuning improves COCO mAP across DETR-family baselines.

### REPRODUCIBILITY STATEMENT

We have taken extensive measures to ensure the reproducibility of our work. Our methodology, algorithms, and theoretical proofs are detailed in Section 3 and Appendix, while the complete experimental setup and all hyperparameters are specified in Section 4. To facilitate direct verification, the supplementary material contains an anonymous, mmdetection-style implementation of our core contributions (the DAP loss and matcher), designed for integration with public Co-DETR repositories. We commit to publicly releasing the full training and evaluation pipeline upon the acceptance of this paper. We are confident that these materials provide a clear path for independent reproduction of our results.

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

# A  APPENDIX

## LLM USAGE DISCLOSURE

In compliance with the conference policy, we disclose the use of a large language model (LLM) as a writing assistance tool during the preparation of this manuscript. The LLM was utilized exclusively for improving grammar, clarity, and readability of the text.

The core research ideas, experimental design, data analysis, and the interpretation of results presented in this paper are entirely the original work of the authors. The LLM was not used for the ideation of research concepts, generation of scientific insights, or production of any data or figures.

The authors have carefully reviewed and edited all content, including any text revised by the LLM, and take full responsibility for the scientific accuracy, integrity, and originality of this work.

## A.1  PARTIAL DERIVATIVE OF AP

To analyze the properties of AP, we compute the partial derivatives of AP with respect to the classification scores. Here, $s_t$ represents the score of a positive sample and $s_n$ represents the score of a negative sample. Note that $s_t$ and $s_n$ independent of the classification threshold $x$. Based on Equation 1, we compute the gradient of the $AP$ with respect to the classification scores.

$$\frac{\partial AP_\alpha}{\partial s_t} = \frac{\int_0^1 P(x)\frac{dR(x)}{dx}dx}{\partial s_t} \tag{15}$$

$$= \frac{\int_0^1 P(x) \times \sum_{b \in \mathcal{T}} f^b(x)dx}{G \times \partial s_t} \tag{16}$$

$$= \frac{1}{G} \times \int_0^1 \frac{\partial P(x)}{\partial s_t} \times \sum_{b \in \mathcal{T}} f^b(x) + P(x)\frac{\partial f^t(x)}{\partial s_t}dx \tag{17}$$

$$= \frac{1}{G} \int_0^1 \frac{\partial f^t(x)}{\partial s_t} P(x) \quad + \tag{18}$$

$$\frac{\frac{\partial Pr(t,x)}{\partial s_t} \times FP(x)}{(T(x) + FP(x))^2} \times \sum_{b \in \mathcal{T}} f^b(x)dx, \tag{19}$$

$$\tag{20}$$

$$\frac{\partial AP_\alpha}{\partial s_n} = \frac{\int_0^1 P(x)\frac{dR(x)}{dx}dx}{\partial s_n} \tag{21}$$

$$= \frac{\int_0^1 P(x) \times \sum_{b \in \mathcal{T}} f^b(x)dx}{G \times \partial s_n} \tag{22}$$

$$= \frac{1}{G} \times \int_0^1 \frac{\partial P(x)}{\partial s_n} \times \sum_{b \in \mathcal{T}} f^b(x) + P(x)\frac{\partial f^t(x)}{\partial s_n}dx \tag{23}$$

$$= \frac{1}{G} \int_0^1 \frac{-\frac{\partial Pr(n,x)}{\partial s_n} \times T(x)}{(T(x) + FP(x))^2} \times \sum_{b \in \mathcal{T}} f^b(x), \tag{24}$$

where $Pr(t,x) = \int_x^1 f^t(y)dy$ represents the probability that the predicted bounding box $t$ is a positive sample given a score threshold $x$, and $\alpha$ is IoU threshold.

## A.2 Partial Derivative of DAP with Respect to Classification Scores

We provide a proof for the proposition mentioned in Section 3.4 of this paper. It is important to note that the DAP presented in this section is equivalent to the AP in Equations 9 and 10. Additionally, the equations in this section consider only cases where $T > 0$. As shown in Equation 10:

$$G \times \frac{\partial AP}{\partial s_n} = \int_0^1 \frac{-\frac{\partial Pr(n,x)}{\partial s_n} \times T(x)}{(T(x) + FP(x))^2} \times \sum_{b \in \mathcal{T}} f^b(x), \tag{25}$$

where, $G$ is the number of gt labels. According to Condition 2, $f(x)$ is greater than 0, and both $T(x)$ and $FP(x)$ are greater than or equal to 0. According to Condition 3, $\frac{\partial Pr(n,x)}{\partial s_n} > 0$. Hence,

$$\frac{\partial AP}{\partial s_n} < 0. \tag{26}$$

As shown in Equation 9:

$$G \times \frac{\partial AP}{\partial s_t} = \int_0^1 \frac{\partial f^t(x)}{\partial s_t} \times P(x) \quad + \tag{27}$$

$$\frac{\frac{\partial Pr(t,x)}{\partial s_t} \times FP(x)}{(T(x) + FP(x))^2} \times \sum_{b \in \mathcal{T}} f^b(x)dx. \tag{28}$$

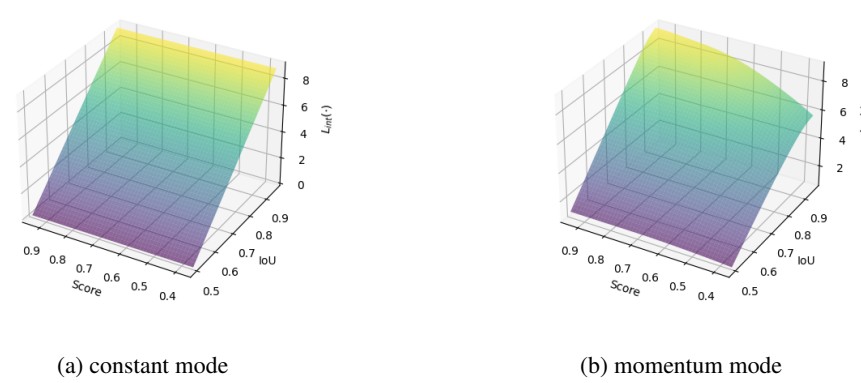

(a) constant mode                           (b) momentum mode

Figure 4: Visualization of $ML_{int}(\cdot)$ in Equation 14

According to Condition 2 and Condition 3 in Section 3.4, $f^b(x) > 0$, so $\frac{\frac{\partial Pr(t,x)}{\partial s_t} \times FP(x)}{(T(x)+FP(x))^2} \times$ $\sum_{b \in \mathcal{T}} f^b(x) > 0$. Therefore,

$$G \times \frac{\partial AP}{\partial s_t} > \int_0^1 \frac{\partial f^t(x)}{\partial s_t} \times P(x)dx, \tag{29}$$

applying integration by parts to the above equation yields:

$$G \times \frac{\partial AP}{\partial s_t} > (P(x) \times \int_0^x \frac{\partial f^t(y)}{\partial s_t}dy)]_0^1 \quad - \tag{30}$$

$$\int_0^1 \frac{\partial P(x)}{\partial x} \int_0^x \frac{\partial f^t(y)}{\partial s_t}dydx \tag{31}$$

$$= -\int_0^1 \frac{\partial P(x)}{\partial x} \int_0^x \frac{\partial f^t(y)}{\partial s_t}dydx \tag{32}$$

$$= -\int_0^1 \frac{\partial P(x)}{\partial x} \frac{\int_0^x f^t(y)dy}{\partial s_t}dx, \tag{33}$$

since we use Interpolated AP, we have $\frac{\partial P(x)}{\partial x} \geq 0$. According to Condition 3 in Section 3.4, we obtain $\frac{\int_0^x f^t(y)dy}{\partial s_t} < 0$. In summary, we have proved that:

$$\frac{\partial AP}{\partial s_t} > 0. \tag{34}$$

Based on Equations 26 and 34, it is evident that our proposed DAP is feasible for optimizing classification tasks.

### A.3 VISUALIZATION OF $ML_{int}(\cdot)$ IN THE MATCHER COST FUNCTION ACROSS DIFFERENT MODES

We set the momentum $m$ in Equation 13 to be close to 1, ensuring that $MP_\alpha^i$ serves as a reasonable approximation of $P_\alpha^i$ on the training set. During the fine-tuning experiments of Co-DETR-SwinL, we visualize $ML_{int}(\cdot)$ in Equation 14. As shown in Figures 4a and 4b, the visualizations of $ML_{int}(\cdot)$ (for scores greater than 0.5) under the momentum mode and the constant mode are remarkably similar in well-trained models. This indicates that the constant mode serves as a reasonable and concise alternative, effective enough to serve as the matching cost function.

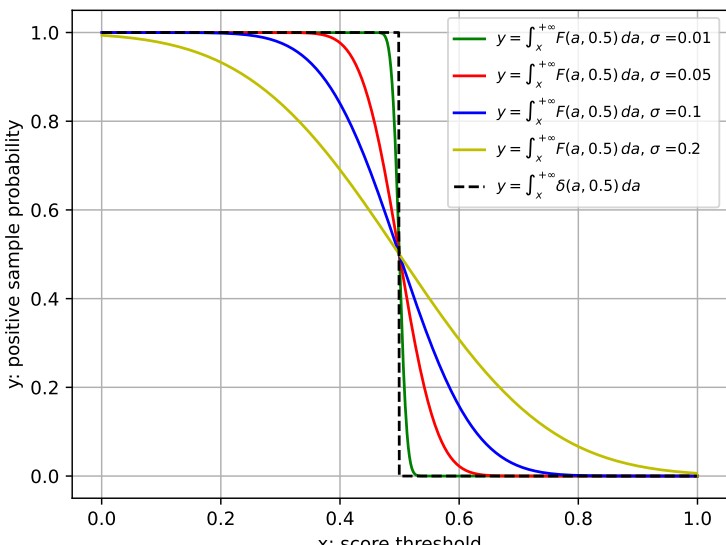

Figure 5: Under a prediction box score of 0.5, the probability of positive samples (y-axis) corresponding to different score thresholds (x-axis). The dashed line represents the function $\delta(\cdot)$ from Equation 3, while the solid line represents the function $F(\cdot)$ from Equation 12.

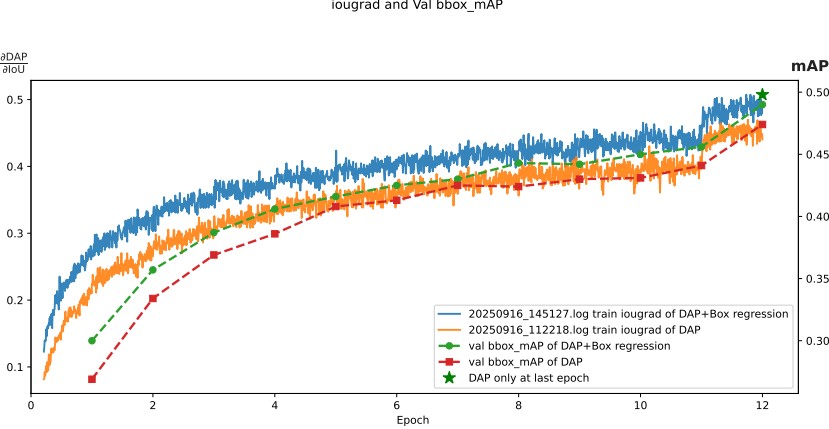

Figure 6: Training dynamics of Co-DETR-R50 from scratch. Blue: IoU gradient magnitude under DAP + box regression; Orange: IoU gradient magnitude under DAP-only; Green: validation mAP under DAP + box regression; Red: validation mAP under DAP-only. Green star markers denote a schedule that uses DAP + box regression for the first 11 epochs and switches to DAP-only in the final epoch.

## A.4  VISUALIZATION OF $F(\cdot)$

As shown in the figure 5, the smaller the standard deviation $\sigma$, the closer $F(\cdot)$ approximates $\chi(\cdot)$. However, as $\sigma$ decreases, the gradient also becomes smaller. Our method achieves optimal performance when $\sigma$ is set to 0.05.

## A.5 ANALYSIS OF TRAINING FROM SCRATCH

As shown in the figure 6, We track localization signal strength via the "IoU grad," defined as the mean absolute gradient of the localization term with respect to IoU(b, gt) over positive matches in a batch (computed from the interpolated localization function $L_int$). The curves are lightly smoothed for visibility. Early in training, DAP-only produces weaker IoU gradients and slower mAP rise, consistent with a flatter precision–recall surface. Adding an auxiliary box regression amplifies IoU gradients and stabilizes optimization, improving mid-epoch accuracy. However, switching to DAP-only in the final stage (green stars: 11 epochs with DAP + box regression, then 1 epoch DAP-only) aligns the objective tightly with COCO-style mAP and yields competitive or better final mAP.

