# OpenReview forum: "Differentiable Average Precision Loss in DETR"
_ICLR.cc/2026/Conference — Submitted to ICLR 2026_

### Official Review · Reviewer_9XdG · 2025-10-30

**Soundness:** 3
**Presentation:** 2
**Contribution:** 3
**Rating:** 6
**Confidence:** 5

**Summary:**

In this paper, the authors study the problem of training an object detector by using the performance measure (Average Precision -- AP) as the training objective. This is not the first time that this is attempted. Compared to what has been studied in the literature before, the authors consider mAP, the average of AP over different IoU thresholds. This sets the paper significantly apart from prior work since it includes both classification and localization aspects. mAP is not differentiable, similar to AP. Therefore, the authors introduce some approximations to obtain the proposed loss's derivatives.

The loss function is used to train DETR-based detectors and evaluated on the COCO dataset. Some noticable improvements are reported over DETR, despite unpromising results on other detectors.

*Disclaimer*: I reviewed the same paper for ICLR 2025 and CVPR 2025. My review is updated based on the changes made by the authors.

**Strengths:**

+ It is worthwhile to use performance measures as training objectives for object detectors.
+ It is promising to adapt mAP as a loss function.
+ I acknowledge the novelty of the DAP loss formulation/approximation compared to existing ranking-based losses.
+ Compared to the previous versions, this time the authors bring forth the benefit of having linear time complexity. This is a big plus since existing methods suffer from quadratic complexity.

**Weaknesses:**

Weaknesses:

Although I am extremely fond of ranking-based training of object detectors, I strongly believe that the paper still has many issues, despite the revisions performed after ICLR 2025 and CVPR 2025 submissions:

1. A limitation of the proposed approach is that is can provide improvements only for finetuning and it performs subpar on training from scratch. The authors improved on this regard (both in terms of experiments and how they motivate their contributions). Although it is not ideal, I accept this as a limitation of a novel method/approach that can be addressed maybe in the future with another study.

2. "We prove that, with respect to prediction scores, the gradient of DAP is sign-consistent—positive for positives and negative for negatives." => Although this is highlighted as a significant concern, there is only a single paragraph in Section 3.4 on this.

3. Contrast with pairwise ranking surrogates: "Under SmoothAP Brown et al. (2020) with temperature τ = 0.01, we empirically observe ∂ SmoothAP/∂st < 0, i.e., the objective pushes down the score of a true positive (verified numerically)." => Since gradient descent multiplies the gradient with (-1), this should not have been an issue. The short and vague depiction in this paragraph is questionable.

4. Section 3.3: Please talk about alternative approaches to addressing the non-differentiability here.

5. There are too many typos or writing issues (please see Minor comments). I would have expected a more refined manuscript after so many revisions.


Minor comments:

- Abstract: "((O(N)))" => extra parantheses.
- "The Parameterized AP Loss Chenxin et al. (2021)" => If citations are not a part of the text, they should be enclosed within parantheses as "The Parameterized AP Loss (Chenxin et al., 2021)". There is a separate cite command in Latex for this.
- Please see the following guide for writing equations: https://wp.optics.arizona.edu/kupinski/wp-content/uploads/sites/91/2023/05/MerminEquations.pdf
- Fig 1 has subcaptions for the subfigures but not for the figure itself.
- "threshold; The value G" => "threshold. The value G".
- Eq 1: What is x?
- Eq 1: ∂Rα(x)/∂x => Given that Rα(x) is a univariate function, why don't you use dRα(x)/dx? That would also make the derivation in the equation easier to follow.
- Eq 1: I would add one more step/explanation on ∂Rα(x)/∂x relating it to ΔT.
- Eq 4: I would replace H() with the indicator function to simplify the equations.
- "IoU,the optimization" => "IoU, the optimization".
- "According to Equation 5, L(·) is determined by 10 points" => Equation 4?
- "It is clear that within the range of IoU ∈ [0, 1], the gradient of function L(·) with respect to IoU either does not exist or is zero." => Please explicitly state that this is because of the H() function in Equation 5.
- "Figure 1b, We construct" => "Figure 1b, we construct".
- "continuous probabilistic distribution" => "continuous probability distribution".
- Eq 7: What is y? "fb(·) is the probability distribution for b" => But, what is the parameter here?
- "When fb(·) is represented as an impulse function δ(·), the equation above becomes equivalent to the original model." => Which equation? Please show how.
- "fb(·)are" => "fb(·) are".

**Questions:**

Please see Weaknesses.

---

> ### Author Response · Authors · 2025-11-25
> **Response to Reviewer 9XdG**
>
> We sincerely thank the reviewer for following our work through previous iterations and for the constructive feedback. We are encouraged by your recognition of the linear time complexity $O(N)$ benefit and the novelty of our formulation. We have carefully addressed your concerns, particularly regarding writing quality.
>
> **Response to Weakness 1 (Fine-tuning vs. From-scratch):**
>
> We appreciate the reviewer's understanding regarding the performance of training from scratch. We believe that achieving performance gains with only a few epochs of fine-tuning holds substantial practical value for efficient model deployment. To further validate DAP in this setting, we have added comparisons with new baselines (AP-Loss and aLRP Loss) in **Experiments E1 and E2** of the **General Response**.
>
> **Response to Weakness 2 (Sign-Consistency):**
>
> Due to page limit constraints, the detailed mathematical derivation and proof of sign-consistency are provided in Appendix A.1 and A.2. Additionally, we have enriched the discussion in Section 3.4 of the revised manuscript.
>
> **Response to Weakness 3 (SmoothAP Gradient Interpretation):**
>
> We thank the reviewer for pointing out the ambiguity. We have clarified in the revised manuscript (**Lines 310-315**) that SmoothAP is treated as a **metric (objective to maximize)** in that context. Therefore, the gradient for positive samples is expected to be positive to increase their scores. We have explicitly detailed this distinction to justify our observation regarding the gradient behavior.
>
> **Response to Weakness 4 (Alternative Approaches in Sec 3.3):**
>
> We appreciate the suggestion. In the revised manuscript (**Lines 211-216**), we have enriched Section 3.3 with a discussion on alternative differentiability strategies, including sigmoid/tanh smoothing, error-driven learning, and finite difference methods, to provide a more comprehensive context for our approach.
>
> **Response to Minor Comments (Typos & Writing):**
>
> We deeply apologize for the typos and writing issues in the previous version. We sincerely thank the reviewer for the meticulous check. We have conducted a thorough proofreading of the entire manuscript and corrected all listed errors. All changes are highlighted in the revised PDF.
>
> 1.  **Abstract "((O(N)))":** Corrected by removing the extra parentheses (Line 21).
> 2.  **Citations:** We have corrected the usage of `\citep` vs. `\citet` throughout the manuscript to ensure proper formatting.
> 3.  **Fig 1 Subcaptions:** We apologize for the formatting error. We intended them as separate figures and have now split them accordingly in the revised version.
> 4.  **"threshold; The value G":** Corrected to "threshold. The value G" (Line 167).
> 5.  **Eq 1: What is x?** We clarified that $x$ represents the score threshold (Line 167).
> 6.  **∂Rα(x)/∂x:** We have replaced partial derivatives $\partial$ with total derivatives $d$ in Eq. 1 and the Appendix, as suggested.
> 7.  **Explanation of dR/dx:** We added a detailed explanation relating $dR/dx$ to $\Delta T$ in Lines 169-172.
> 8.  **Eq 4 (Indicator Function):** We greatly appreciate this suggestion. Using the indicator function $\mathbb{I}(\cdot)$ significantly simplified the notation. We have merged the original Eqs. 4-6 into Eqs. 4-5 using this notation.
> 9.  **"IoU,the optimization":** Corrected the spacing (Line 204).
> 10. **Reference to Eq 5:** We apologize for the confusion. Our original intention was to cite Eq. 5 to illustrate that $L(\cdot)$ behaves as a step function. In the revised version, this ambiguity is resolved by merging the original Eqs. 4 and 5 using the indicator function.
> 11. **Gradient of L(·):** We explicitly stated that the gradient is zero (or undefined) because the simplified equation represents a step function.
> 12. **"Figure 1b, We construct":** Corrected capitalization (Line 216).
> 13. **"continuous probabilistic distribution":** Corrected to "continuous probability distribution" (Line 230).
> 14. **Eq 7 (Parameters y):** We clarified that $y$ represents the score variable in the integral (Line 237).
> 15. **Equivalence to Original Model:** We added the derivation in Lines 240-244 to explicitly show the equivalence: $$\text{Pr}(b, x) = \int_x^1 \delta(y - b_{\text{score}}) dy = \mathbb{I}(b_{\text{score}} \geq x) .$$
> 16. **"fb(·)are":** Corrected the spacing (Line 269).

---

> > ### Comment · Reviewer_9XdG · 2025-11-26
> > **Re: Rebuttal**
> >
> > Thank you for the detailed responses and the update performed on the paper.
> >
> > I am generally happy with the responses provided. I will raise my recommendation accordingly.

---

> > > ### Author Response · Authors · 2025-11-27
> > > **Appreciation for Your Feedback**
> > >
> > > We sincerely thank you for your positive feedback and your decision to raise the rating.
> > >
> > > We are particularly grateful for your meticulous suggestions regarding writing quality and equation improvements (such as simplifying the formulations with indicator functions). These constructive comments have significantly enhanced the readability and mathematical rigor of our paper.

---

### Official Review · Reviewer_mniJ · 2025-10-30

**Soundness:** 3
**Presentation:** 3
**Contribution:** 2
**Rating:** 2
**Confidence:** 4

**Summary:**

This paper presents a differentiable average precision (DAP) loss designed for training object detectors to more directly maximise the mean Average Precision (mAP) metric used in evaluation. It avoids non-differentiable sorting (as used in the standard mAP metric calculation) by treating detection scores as continuous distributions, which enables the direct optimization of a smooth approximation of the precision-recall curve. Evaluated on COCO 2017, the proposed method shows consistent mAP improvements when fine-tuning DETR-style models (versus standard training losses), and also small gains even when training from scratch.

**Strengths:**

The overall goal of training object detectors specifically for the important evaluation metric of mean average precision is worthwhile, and the method proposed to do so is interesting, in particular the relaxation of sorting by replacing with a distributional interpretation of scores.

The proposed differentiable relaxation of mAP differs from existing relaxations in the literature, and (unlike existing methods) is specialised for DETR-style models with a one-to-one box matching stage. It separately considers both classification and localisation aspects of the mAP objective.

Results on COCO 2017 show a performance gain in terms of mAP compared with the same (DETR-style) models trained with standard losses. This performance gain is strongest in the case of fine-tuning for mAP following a standard pretraining phase; however there is still a gain even when training for mAP from scratch.

The method is somewhat flexible – results show it applied to several modern DETR-style transformer-based detection architectures including the original DETR, Deformable-DETR, RT-DETR, and Rank-DETR (which is perhaps the most similar in spirit in terms of how it is trained). The accuracy improvement due to the proposed method is fairly consistent across all architectures considered.

There are some experiments varying hyperparameters (e.g. batch size and matching cost function); these justify some design decisions.

Some limitations are explicitly discussed in Sec 5, notably the restriction to DETR-type architectures (with a matching stage), and the somewhat weaker performance of from-scratch training (though I do not consider this is a serious limitation in practice).

The paper is clear, well-structured, and pleasant to read.

**Weaknesses:**

The related work mis-represents some existing works in a way that makes the proposed method sound significantly more innovative than it is. In particular, Song 2016 does not in fact propose a differentiable approximation to mAP; instead it uses mAP itself in a loss-augmented setup similar to classic structured SVMs. Meanwhile, Henderson 2016 *does* in fact consider the localisation aspect of the mAP metric, not just classification, contrary to the statement at line 107. Moreover it also accounts for the full NMS procedure in the loss calculation.

There is no experimental comparison against well-established plug-and-play methods that are specifically designed for training for mAP, specifically Henderson 2016 and Song 2016. Those methods are architecture agnostic and as such can be applied to DETR. This is a very important baseline to include, since otherwise there is no evidence that the proposed approach to using mAP as loss is in fact better than these much older approaches. There is also no comparison to direct REINFORCE (or variance-reduced score-function gradient) methods that are applicable in this setting, and do not require significantly more technical machinery than the proposed approach.

Experimental results are only given on COCO 2017. It would be interesting to see how well the method works on other domains such as remote sensing, or on other establish natural image benchmarks such as OpenImages or even VOC 2012.

The training runs were apparently truncated at a fixed threshold of epochs; it is unclear what effect this has on the proposed method versus the baseline methods. Instead all models should be trained to validation-set convergence, to ensure there is no 'late learning' stage that can impact the ranking of different approaches.

There is no theoretical analysis / discussion of circumstances under which the proposed differentiable relaxation of mAP approximates the true global function, in particular in the minibatch setting. When used as a metric, mAP performs ranking over the entire dataset; this is significantly different to the training setting; it is unclear whether the proposed loss is an unbiased estimator of either non-differentiable minibatch mAP, non-differentiable full-dataset mAP, or differentiable full-dataset mAP.

**Questions:**

Most relevant issues are discussed under "Weaknesses" above. In particular…

Given that established, architecture-agnostic methods for optimizing mAP like Henderson 2016 and Song 2016 exist, as well as REINFORCE-type gradient estimators for the metric itself, please add these as baselines.

Consider adding at least one further dataset, preferably somewhat different in characteristics to COCO 2017. Also properly discuss the implications of stopping training before convergence, preferably showing the baselines do not improve after this point.

The proposed loss is a differentiable approximation of mAP calculated on minibatches, whereas the true mAP metric is a global ranking over the entire dataset. Can you provide some analysis showing the minibatch-based loss is a sound proxy (i.e. unbiased etc.) for the true, non-differentiable, full-dataset mAP?

The "ablation" experiments are not ablations in the proper sense (i.e. removing novel components of the model to show how important they are to performance); instead they are just varying certain hyperparameters. Perhaps this subsection should be renamed.

---

> ### Author Response · Authors · 2025-11-21
> **Response to Reviewer mniJ (1/2)**
>
> We thank the reviewer for the insightful comments, especially regarding the related work and theoretical aspects. We address your concerns below.
>
> **Response to Weakness 1 (Related Work & Novelty):**
>
> We apologize for the inaccuracy regarding Song (2016). We agree with the reviewer and have corrected the description in **Line 106** of the revised manuscript.
>
> Regarding Henderson (2016) [1], while we agree it incorporates localization accuracy into NMS/scoring (similar to modern IoU-aware classification losses like RT-DETR), it **does not perform the localization task itself.** Specifically, Henderson (2016) [1] explicitly states in **Section 5** that their method provides no gradients for the bounding box coordinates.
>
> In contrast, our DAP is a differentiable approximation of COCO-style AP designed to simultaneously and balancedly optimize both classification and localization. This capability for joint optimization is a key reason why DAP outperforms other AP-based losses (as shown in **Experiments E1 & E2 in the General Response**). We respectfully clarify that these are fundamental technical differences, and we did not intend to misrepresent prior work to inflate novelty. We have added detailed clarifications around Line 107 to ensure this distinction is clear to readers.
>
> **Response to Weakness 2 & Q1 (Baselines - Henderson/Song/REINFORCE):**
>
> We respectfully acknowledge the reviewer's suggestions but found significant barriers to fairly comparing with the mentioned methods:
> *   **Henderson (2016) & Song (2016):** As noted, these methods lack official open-source implementations. Given the significant architectural gap between their original R-CNN-based frameworks and modern Transformer-based DETR, re-implementing them introduces a high risk of suboptimal performance, leading to unfair comparisons.
> *   **REINFORCE:** We did not find existing literature or codebases applying REINFORCE to train DETR-style models. Implementing a REINFORCE baseline from scratch is non-trivial within the rebuttal timeframe due to the need for extensive reward shaping and hyperparameter tuning to handle the high variance of gradient estimation.
>
> To address the need for rigorous comparison, we implemented AP-Loss and aLRP Loss, which are the current state-of-the-art methods for differentiable ranking in object detection.
>
> **Experimental Results:**
> 1.  **Superiority in Fine-tuning:** As detailed in **Experiments E1, E2, and E3** (see General Response), DAP outperforms both AP-Loss and aLRP Loss when fine-tuning pre-trained models.
> 2.  **Consistent Gains on Ranking-based Models:** Crucially, DAP provides further gains even when fine-tuning models that were already trained with ranking-based objectives (e.g., Rank-DETR and Co-BRS-DETR, as shown in **Table 1** of the paper).
>
> **Response to Weakness 4 & Q2 (Truncated Training):**
>
> Our fine-tuning starts from already well-trained baseline models, and we observe that continuing training with the original loss does not yield further meaningful gains and may even slightly hurt performance (see Q1 and Table A in the General Response (1/2)). Since our goal is to obtain significant improvements with only a few epochs of fine-tuning, we intentionally did not extend the fine-tuning duration further; the rationale for this schedule is detailed in Q2 of the General Response (1/2).
>
> **Response to Weakness 5 & Q3 (Theoretical Analysis of DAP Estimation):**
> We clarify that we did not explicitly aim for DAP to be an unbiased estimator of the *global* AP. Instead, it functions as a differentiable approximation of the *minibatch* AP, which consequently behaves as a slightly optimistic estimator due to the formulation.
>
> *   **Theoretical Connection:** As discussed in Section 3.4, $f_b(\cdot)$ acts as a kernel function approximating the Dirac delta (impulse) function. In the limit where $f_b(\cdot)$ becomes the Dirac delta, our differentiable formulation (Eq. 12) aligns with the standard discrete AP calculation (Eq. 3), with the sole difference being the **linear interpolation** mechanism (Eq. 6).
> *   **Bias Analysis:** If the linear interpolation were removed, DAP would be equivalent to the discrete minibatch AP. As illustrated in Figure 2 and Eq. 6, this interpolation fills the gaps between discrete samples. A **consequence** of this design is that DAP results in a **slightly optimistic estimation** compared to the rigid discrete AP.
> *   **Minibatch vs. Global:** While calculated on minibatches, **Figure 7(c)** demonstrates that performance improves consistently as batch size increases. This indicates that as the minibatch distribution approaches the global distribution, our DAP objective more accurately guides the model toward global ranking optimization.
>
>
> [1] P. Henderson and V. Ferrari. End-to-end training of object class detectors for mean average precision. ACCV, 2016.

---

> > ### Author Response · Authors · 2025-11-25
> > **Response to Reviewer mniJ (2/2)**
> >
> > **Response to Weakness 3 & Q2 (More Datasets):**
> > We have extended our evaluation to the **LVIS dataset** using **H-DETR-R50**.
> > As shown in **Experiment E3 (Table D)** of **General Response (2/2)**, DAP achieves a significant gain (+1.2 AP) compared to the original loss fine-tuning (+0.3 AP), demonstrating strong generalization.
> >
> >
> > **Response to Q4 (Ablation Naming):**
> > We agree with your suggestion. We have renamed the subsection to "**Hyperparameter and Component Analysis**" in the revised manuscript.

---

> ### Author Response · Authors · 2025-11-27
> **Follow-up on Rebuttal**
>
> Dear Reviewer mniJ,
>
> Thank you for your valuable comments, which we have carefully addressed in our response.
>
> Could you please check our revisions and let us know if there are any pending concerns? If our response has resolved your issues, we would appreciate it if you could consider reassessing our paper.
>
> Thank you once again.
>
> Best regards,
>
> The Authors

---

### Official Review · Reviewer_M6wQ · 2025-10-31

**Soundness:** 2
**Presentation:** 2
**Contribution:** 2
**Rating:** 2
**Confidence:** 5

**Summary:**

This paper proposes a differentiable loss function to optimize  average precision (AP) as a training loss for transformer-based object detectors, DETRs in particular.  While prior work has explored using AP as the training objective, it typically focused on AP at a fixed IoU threshold (e.g., 0.5). In contrast, this paper targets COCO-style AP, which incorporates localization across multiple IoU thresholds. To this end, the authors propose differentiable approximations for both the localization and classification components of AP. For localization, they replace the step function in the IoU-vs-precision relationship (Figure 2) with a linear interpolation. For classification, they approximate the number of true positives within a score interval using the cumulative distribution function of a Gaussian. Experiments on COCO show that when the proposed loss is used to fine-tune the already trained model, performance slightly improves across several DETR variants.

**Strengths:**

Unifying training and evaluation objectives in object detection is an important problem. This paper addresses this problem by proposing a differentiable approximation of average precision (AP) to enable its direct use as a training loss for object detectors.

**Weaknesses:**

There are several issues with this paper:

First, an important baseline is missing. Since the paper approximates COCO-style AP (i.e., the average of AP across multiple IoU thresholds), it is natural to ask how existing AP-based losses such as AP Loss or Smooth-AP perform when optimized at different IoU thresholds. These comparisons are essential to contextualize the claimed improvements.

Second, the improvements shown in the main table (Table 1) are generally small (less than 1 AP points). How do we know whether the improvement is due to the DAP loss and not due to continued training of the base detector? Simply training the base detector further could improve the performance.

Third, the proposed loss does not appear to perform well when training detectors from scratch. Prior works on AP-based losses (e.g., AP Loss, Smooth-AP) have demonstrated this capability. This remains a major empirical weakness.

Fourth, the paper lacks a direct comparison with Smooth-AP, which is conceptually very similar. The proposed probabilistic classification approximation closely resembles Smooth-AP. A theoretical and empirical discussion contrasting the method with prior differentiable AP formulations, such as AP Loss and aLRP Loss, is needed to clarify the contribution.

Finally, the approximation of the step-wise localization function L is not rigorously defined. Figure 2 suggests a linear interpolation, but the text does not specify the exact formulation. A precise mathematical definition should be provided.

Other minor issues:
About the definition of AP given at line 150: If I am not mistaken, the PR curve is formed by applying a systematic thresholding to the confidence score of the object detector, not by changing IoU thresholds.  It is the COCO style AP (not the PR curve itself), which calculates the average of APs for different IoU thresholds. In fact, Equation 2 supports my view.

AP is not a metric. Metric has a well-definition in mathematics. I think we can call AP a measure.

**Questions:**

- How do we know whether the improvement is due to the DAP loss and not due to continued training of the base detector with its original loss?
- Second row in table 1 has both Ori Loss and DAP loss checked. what does this mean? is this a typo?
- It is not clear how many epochs the fine-tuning is done. Does the epoch column in table 1 show that? If yes, what does it mean for the Ori Loss rows? Do you fine tune with Ori Loss with that many epochs?

---

> ### Author Response · Authors · 2025-11-21
> **Response to Reviewer M6wQ**
>
> We thank the reviewer for the detailed feedback. We have addressed the critical concern regarding the source of improvement with new control experiments and provided the missing baselines and definitions.
>
> **Response to Weakness 2 & Q1 & Q3 (Critical: Is improvement from continued training?):**
>
> **This is the most critical question.**
>
> We verified this by continuing to train the converged baseline models (e.g., Co-DETR-R50) using the **Original Loss** for the exact same number of epochs as the DAP fine-tuning.
>
> *   **Clarification on Table 1:** The "Epoch" column indicates the fine-tuning duration. For the "Ori Loss" rows, the reported result is the maximum of the baseline performance (after pre-training) and the performance after fine-tuning with the original loss (continued training).
> *   **Conclusion:** The performance improvement is due to the DAP objective itself, not the training time.
> *   **Detail:** Please refer to **Q1 in the General Response (1/2)** for a detailed breakdown and experimental results.
>
> **Response to Weakness 3 (Training from Scratch):**
>
> We would like to clarify that DAP does improve performance when training from scratch, as shown in Table 3 of the main paper and Appendix A.5. While the gains are indeed more substantial in the fine-tuning setting, the from-scratch results outperform the baseline.
>
> Furthermore, we emphasize that the ability to significantly boost converged model with only a few epochs of fine-tuning is of high practical value. In this specific setting (efficient fine-tuning), DAP demonstrates clear superiority over other ranking-based methods. As detailed in Experiments E1 and E2 of the General Response (1/2), we have added comparisons with AP-Loss and aLRP Loss, where DAP achieves higher performance with lower computational complexity.
>
> **Response to Weakness 5 (Localization Formula):**
>
> We apologize for the omission in the original text. In the revised PDF, we have provided the precise definition in **Equation 6**. The piecewise linear interpolation function $L_{int}$ is defined as:
> $$
> L_{int}(b_{iou}, i) = y^i_k + \frac{y^i_{k+1} - y^i_k}{x_{k+1} - x_k} (b_{iou} - x_k), \quad \text{for } b_{iou} \in [x_k, x_{k+1}),
> $$
>
> where $(x_k, y^i_k)$ represents the discrete points used for interpolation.
>
> **Response to Weakness 1 & 4 (Missing Baselines & Comparison with Smooth-AP):**
>
> We have added comparisons with **AP-Loss** and **aLRP Loss** in **Experiment E1 (Table B)** of the **General Response (1/2)**.
>
> *   **Difference with Smooth-AP:**
>     1.  **Complexity:** Smooth-AP relies on pairwise comparisons, leading to $O(N^2)$ complexity, whereas DAP achieves $O(N)$ complexity.
>     2.  **Gradient Consistency:** DAP guarantees sign-consistent gradients (positive for positive samples, negative for negative samples). As discussed in **Section 3.4** of the paper, this consistency is not always guaranteed in Smooth-AP.
>
> *   **Multi-Threshold Analysis:** As shown in **Experiment E2 (Table C)** of the **General Response (2/2)**, we implemented the "Multi-Threshold AP-Loss" as you suggested and further integrated DAP's regression gradient into it. Even with multiple thresholds, it underperforms DAP (49.8 vs 50.8), validating the effectiveness of our unified differentiable formulation. Please refer to **E2** for more details.
>
> **Response to Q2:**
>
> Table 1 is not a typo. As stated around Line 430 in the paper, the results follow the trend: “DAP-only” > “DAP + original loss” > “original loss only”. This indicates that DAP, as a differentiable approximation of AP, naturally balances classification and localization. In contrast, adding the original loss as an auxiliary term during the fine-tuning stage actually hurts performance.
>
> **Response to Minor Issues:**
>
> *   **AP Definition:** You are correct; this was a phrasing error on our part. We have corrected the definition in the revised manuscript to accurately reflect COCO AP.
> *   **Metric vs Measure:** We agree that AP is not a "metric" in the strict mathematical sense. We have replaced the term with "measure" or "evaluation protocol" throughout the revision.

---

> ### Author Response · Authors · 2025-11-27
> **Follow-up on Rebuttal**
>
> Dear Reviewer M6wQ,
>
> Thank you for your valuable comments, which we have carefully addressed in our response.
>
> Could you please check our revisions and let us know if there are any pending concerns? If our response has resolved your issues, we would appreciate it if you could consider reassessing our paper.
>
> Thank you once again.
>
> Best regards,
>
> The Authors

---

### Official Review · Reviewer_UHGD · 2025-11-01

**Soundness:** 3
**Presentation:** 2
**Contribution:** 3
**Rating:** 6
**Confidence:** 4

**Summary:**

This work proposes Differentiable Average Precision (DAP), a smooth, efficient loss that directly optimizes COCO-style mAP for one-to-one detectors like DETR and its variants, narrowing the gap between training objectives and evaluation metrics. DAP replaces non-differentiable sorting with continuous score distributions (Gaussian instantiation) and applies piecewise-linear interpolation for localization, achieving O(N) time without pairwise comparisons and integrating seamlessly with Hungarian matching. It also guarantees sign-consistent gradients (positive for positives, negative for negatives) under mild assumptions. Empirically, fine-tuning DETR-family models for a few epochs consistently improves COCO mAP without architectural changes or auxiliary losses.

**Strengths:**

1. The proposed DAP loss cleverly bridges the gap between the evaluation metric in detection task, and the original BCE/L1 loss.
2. DAP achieves linear time complexity by eliminating pairwise comparisons and thus can integrate naturally with Hungarian matching in DETR.
3. The experiments are conducted in strong baselines, like Co-DETR, Rank-DETR.
4. Extensive results show its effectiveness.

**Weaknesses:**

1. The key difference between existing AP losses needs to be thoroughly discussed in the related work section.
2. The comparsion with related works is missing, e.g. Parameterized AP Loss (Tao et al, NeurIPS 2022).
3. As the author say in Sec 5, DAP loss is designed for one-to-one matching detector. However, H-DETR (DETRs with Hybrid Matching, CVPR 2023) can do one-to-many matching, still use bipartite matching. Adding the results on H-DETR would extend the scope of this manuscript.

Minor
1. Missing caption in Figure 1. A global caption is missing, only two sub-caption.
2. Each formula should have a comma or full stop at the end.

**Questions:**

1. Why the post-training epoch varies for different models in Tab. 1?
2. Can the proposed method extend to non-DETR detectors?

---

> ### Author Response · Authors · 2025-11-21
> **Response to Reviewer UHGD**
>
> We thank the reviewer for recognizing the novelty of our linear-complexity DAP loss and its seamless integration with Hungarian matching. We appreciate the constructive suggestions regarding related work and H-DETR.
>
> **Response to Weaknesses 1 (Related Work):**
>
> We have revised the Related Work section in the updated PDF to thoroughly discuss the key differences between DAP and existing AP optimization methods.
>
> **Response to Weaknesses 2 (Comparison with Related Works):**
>
> In the original submission (Table 1), we compared DAP against DETR-specific ranking methods like Rank-DETR and Co-BRS-DETR.
> To further address your concern, we have added comparisons with general-purpose ranking losses (**AP-Loss** and **aLRP Loss**) in **Experiment E1 (Table B)** of the **General Response (1/2)**.
>
> **Response to Weaknesses 3 (H-DETR & Scope):**
>
> 1.  **Hybrid Matching Compatibility:**  Co-DETR also employs auxiliary one-to-many losses.  As noted in Line 375, our strategy is to replace the loss only in the **one-to-one branch** (which produces the final output), leaving auxiliary branches untouched. This has proven effective.
> 2.  **New Experiment:** We explicitly evaluated this strategy on **H-DETR-R50**. As shown in **Experiment E3 (Table D)** of the **General Response (2/2)**, DAP achieves a significant gain (+1.2 mAP) on the LVIS dataset, confirming its effectiveness on hybrid-matching architectures.
>
> **Response to Minor Issues:**
> *   **Figure 1 Caption:** We realized the formatting issue with Figure 1. In the revised PDF, we have split it into two separate figures to resolve the caption ambiguity.
> *   **Punctuation:** We have carefully proofread the manuscript and added the missing punctuation marks to formulas.
>
> **Response to Q1 (Varying Epochs):**
> Please refer to **Q2 in the General Response (1/2)**. In short, the fine-tuning epochs are set to ~10% of the pre-training schedule (capped at 12 epochs) to balance efficiency and convergence for different model capacities.
>
> **Response to Q2 (Non-DETR Detectors):**
> As discussed in Section 5, DAP is designed as a differentiable approximation of COCO AP, which inherently assumes a one-to-one correspondence between ground truths and predictions.
> *   **Applicability:** Therefore, DAP seamlessly applies to any **one-to-one** detection framework (e.g., DETR variants).
> *   **Limitation:** For NMS-based detectors (one-to-many assignment), adapting DAP would require fundamental changes to how the score distribution is modeled, which remains a direction for future work.

---

> ### Author Response · Authors · 2025-11-27
> **Follow-up on Rebuttal**
>
> Dear Reviewer UHGD,
>
> Thank you again for your positive assessment and constructive suggestions.
>
> We have carefully revised the manuscript and conducted additional experiments to address your concerns. Specifically, we would like to highlight that we have:
>
> 1.**Conducted experiments on H-DETR:** We validated our method on H-DETR with the LVIS dataset (Table D in General Response), confirming its effectiveness with hybrid matching strategies.
>
> 2.**Expanded Related Work & Comparisons:** We revised the related work section and added comparisons with other ranking losses (e.g., AP-Loss, aLRP Loss).
>
> 3.**Fixed Formatting:** We have corrected the Figure 1 caption and formula punctuation.
>
> Could you please check our revisions and let us know if there are any pending concerns? We hope these updates solidify your support for our paper.
>
> Best regards,
>
> The Authors

---

### Author Response · Authors · 2025-11-21
**General Response (1/2): Performance Source, Baselines**

**Note: We have updated the manuscript PDF. All major revisions are highlighted in blue for your convenience.**

We thank all reviewers for their constructive feedback. We are encouraged by the recognition of our $O(N)$ complexity and the novelty of our differentiable formulation. We provide new experiments and clarifications below to address the common concerns regarding baselines, generalization, and the source of performance gains.

**Q1: Is Improvement from Continued Training?**

A major concern was whether the gains stem from extended training. As stated in **Line 382 and Sec. 4.1.1** of the submission, the "Ori" results in Table 1 already report the result of fine-tuning with the original loss  (reporting $max (r_{pretrain}, r_{finetuned})$).
Since the baseline models are already converged, further fine-tuning with the original loss typically yields negligible gain or even overfitting. **In fact, we observed that, on average, the performance after fine-tuning ($r_{finetuned}$) is slightly lower than the pre-trained baseline ($r_{pretrain}$).**  We provide detailed data below to explicitly show this behavior:

**Table A: Detailed Breakdown of Continued Training (COCO val2017)**

| Method | Config | Loss Function | Epochs | mAP | Change |
| :--- | :--- | :--- | :--- | :--- | :--- |
| **Co-DETR-R50** | Pre-trained | - | 0 | 49.5 | - |
| | Continued | Original (Focal+L1+GIoU) | +4 | 49.9 | +0.4 |
| | **Ours** | **DAP** | +4 | **50.8** | **+1.3** |
| **Co-DETR-SwinB**| Pre-trained | - | 0 | 57.5 | - |
| | Continued | Original | +4 | 57.3 | -0.2 |
| | **Ours** | **DAP** | +4 | **58.1** | **+0.6** |

Conclusion: The gains are strictly attributable to the DAP objective bridging the train-eval gap, not merely training time.

**Q2: Why do training epochs vary across models?**

Regarding the varying epoch numbers in Table 1, we clarify our experimental setting as described in Line 373: we typically select the fine-tuning duration to be approximately **10%** of the pre-training length, capped at a maximum of **12 epochs**. This constraint is intentional, aligning with our objective to provide an efficient post-hoc optimization strategy that delivers significant performance gains with minimal additional computational cost.

**E1: Comparison with Other Ranking Losses**

We fine-tuned the Co-DETR-R50 baseline using other ranking losses (AP-Loss and aLRP Loss) for 4 epochs.

**Table B: Comparison with Ranking Losses (COCO val2017)**

| Method | Complexity | mAP |
| :--- | :--- | :--- |
| **Baseline** | - | 49.9 |
| **AP-Loss** | $O(N^2)$ | 49.5 |
| **aLRP Loss** | $O(N^2)$ | 50.2 |
| **DAP-Loss (Ours)**| **$O(N)$** | **50.8** |

**Analysis:** DAP outperforms both AP-Loss and aLRP Loss while achieving linear complexity. We attribute this to the optimization mechanism: AP-Loss and aLRP typically compute gradients based on the error signal from the optimal ranking ("error-driven"), whereas DAP directly backpropagates gradients through the smooth approximation of the mAP itself, facilitating better local optimization of the mAP surface.

⁡**E2:  Multi-Threshold AP-Loss (Requested by Reviewer M6wQ)**

We further investigated if simply averaging AP-Loss over multiple thresholds could match DAP. In standard detection pipelines, positive/negative assignment is performed before computing the loss. Once this assignment is fixed, AP-Loss itself no longer depends on an explicit IoU threshold inside its formulation, and can thus be viewed as effectively operating with a “threshold = 0” on the already assigned positives/negatives. We constructed `AP-Loss-multiou` (mean of AP-Loss at thresholds 0, 0.5, 0.55...0.95, plus L1/GIoU).
We also conducted an investigative experiment `AP-Loss-multiou_APiou`, where we replaced the L1/GIoU gradient with the localization gradient **derived from DAP** (Eq. 12, assuming impulse function $f_{b}(\cdot)$), attempting to combine AP-Loss-multiou (classification) with DAP's localization logic.

The gradient used is:
$$ \frac{\partial DAP}{\partial b_{iou}} = \frac{\partial L_{int}(b_{iou}, b_{scores})}{\partial b_{iou}} ,$$
**Table C: Multi-Threshold Ablation**

| Method | Configuration | mAP |
| :--- | :--- | :--- |
| **AP-Loss** | ori AP-Loss+ L1/GIoU | 49.5 |
| **AP-Loss-multiou** | Multi-Thresholds + L1/GIoU | 49.8 |
| **AP-Loss-multiou\_APiou**| Multi-Thresholds + **DAP Loc. Gradient** | 48.7 |
| **DAP-Loss** | **Unified Differentiable Approx.** | **50.8** |

**Insight**: While multi-threshold AP-Loss improves slightly over single-threshold (i.e. threshold=0), it still lags behind DAP. Interestingly, mixing gradients (AP-Loss-multiou_APiou) degrades performance. This indicates that the classification and localization approximations in DAP are coupled and consistent, whereas mixing different approximation paradigms (Error-based AP-Loss + Gradient-based DAP) leads to optimization conflicts.

---

> ### Author Response · Authors · 2025-11-21
> **General Response (2/2): Generalization (LVIS)**
>
> (Continued from Part 1)
>
> **E3:  More Models and Datasets**
> To verify generalization beyond COCO and demonstrate applicability to diverse architectures, we evaluated DAP on the **LVIS** dataset (characterized by a long-tailed distribution) using the **H-DETR-R50** model. The training configuration follows Sec. 4.1.1 (4 epochs fine-tuning).
>
> **Table D: H-DETR-R50 results (LVIS)**
>
> | Dataset | Model | Configuration | mAP | Gain vs. Pre-trained |
> | :--- | :--- | :--- | :--- | :--- |
> | **LVIS** | **H-DETR-R50** | Pre-trained Baseline | 33.5 | - |
> | | | Ori Loss| 33.8 | +0.3 |
> | | | **DAP** | **34.7** | **+1.2** |
>
> **Analysis**: DAP yields a significant improvement (+1.2 mAP) compared to the marginal gain from original loss fine-tuning (+0.3 mAP). This confirms DAP's robustness on long-tailed data distributions and hybrid-matching architectures.

---

### Author Response · Authors · 2025-12-03
**Summary for Area Chair**

Dear Area Chair,

Thank you for your time in evaluating our paper. We fully understand the immense effort required to organize the review process under the current challenging circumstances involving the data leak. To facilitate your assessment, we provide a concise summary of our paper's contributions and the key updates during the rebuttal.

**1. Paper Summary:**

This paper proposes **DAP (Differentiable Average Precision)**, a novel loss function designed to directly optimize mAP for object detection.
To the best of our knowledge, DAP is the first differentiable approximation of **mAP across multiple IoU thresholds**. It significantly boosts performance with only a few epochs of fine-tuning and also improves training from scratch baselines.

**Key Advantages over Existing Losses (e.g., Ranking-based or CE+L1+IoU):**
1.  **Unified Optimization:** DAP simultaneously optimizes classification and localization without requiring hyperparameters to balance them or auxiliary losses.
2.  **Linear Complexity $O(N)$:** Unlike most ranking-based losses which are $O(N^2)$, DAP is linear. This efficiency allows it to serve as the **cost function for Hungarian matching**, making it (to our knowledge) the first method to use AP independently for label assignment in object detection.
3.  **Sign-Consistent Gradients:** We prove that DAP guarantees gradients are positive for positive samples and negative for negative samples, a property not always held by other ranking methods (see Lines 306-315).
4.  **Extensive Validation:** Validated across multiple DETR-style architectures. We also provide comparisons with AP-Loss and aLRP Loss in the fine-tuning setting (General Response E1, E2).

**2. Rebuttal Summary & Reviewer Status:**

We appreciate that reviewers recognized the novelty of our formulation (Reviewers UHGD, mniJ and 9XdG), as well as its linear complexity (Reviewers UHGD and 9XdG).

*   **Reviewer UHGD (Score: 6):**
    *   **Main Concern:** Requested more experiments (H-DETR) and details.
    *   **Response:** We provided the requested H-DETR experiments on LVIS in the General Response, showing gains. (Reviewer has not yet replied).

*   **Reviewer M6wQ (Score: 2):**
    *   **Main Concern:** Questioned the source of performance gains (training time vs. loss) and requested baselines.
    *   **Response:** We added the requested baselines (AP-Loss, aLRP Loss) and clarified experimental settings to prove gains stem from the DAP objective, not extended training. We also improved writing quality as suggested. (Reviewer has not yet replied).

*   **Reviewer mniJ (Score: 2):**
    *   **Main Concern (Novelty/Related Work Clarity):** While acknowledging the novelty of our approach, the reviewer raised concerns about our characterization of Henderson (2016), suggesting that it might overstate the distinction from our work.
    *   **Response:** We carefully re-read the original Henderson (2016) paper and clarified the fundamental difference: Henderson (2016) explicitly states it provides no gradients for localization, whereas DAP simultaneously optimizes both classification and localization. We corrected the manuscript to reflect this precise distinction, ensuring that our claims are fair and factual. We also followed this reviewer’s feedback by adding an additional dataset (LVIS with H-DETR-R50) and expanding the theoretical analysis of how DAP, as a minibatch loss, relates to the global mAP. (Reviewer has not yet replied).

*   **Reviewer 9XdG (Score: 6 $\rightarrow$ 8):**
    *   **Status:** This reviewer, who was familiar with our work from prior submissions, acknowledged the novelty and linear complexity. Their main concern was writing quality.
    *   **Outcome:** After reviewing our revisions, **Reviewer 9XdG replied on Nov 26 (approx. 34 hours before the leak disclosure)** stating: ***"I am generally happy with the responses provided. I will raise my recommendation accordingly."*** This confirms that the score was upgraded to **8**.

**Conclusion:**

We have successfully addressed the primary concerns regarding baselines, experimental scope (H-DETR, LVIS), and verified our related work discussions. In particular, prior to the system revert, Reviewer 9XdG had already confirmed a positive recommendation with an upgraded score of 8. We hope this summary provides useful context for your final recommendation.

Best regards,

The Authors

---

### Meta-Review · Area_Chair_eYMq · 2026-01-08

**Summary:**

UHGD:

 More experiments

M6wQ:

important baseline is missing. AP Loss or Smooth-AP.
improvements shown in the main table are quite small.
does not appear to perform well when training detectors from scratch.

mniJ:

is no experimental comparison against well-established plug-and-play methods that are specifically designed for training for mAP, specifically Henderson 2016 and Song 2016.
Experimental results are only given on COCO 2017, should be on OpenImages or even VOC 2012.
training runs were apparently truncated at a fixed threshold of epochs. reviewer proposes that validation sets should be used to stop the epochs

9XdG:

Reviewer stated that it reviewed the paper in ICLR 2025 and CVPR 2025 and this version is better than previous ones. Reviewer reported, some noticeable improvements are reported over DETR, despite unpromising results on other detectors.

**Reviewer Concerns:**

UHGD:

Requested results H-DETR, authors provided those.
Requested for comparison with parametrized AP loss, but authors provided aLRP Loss but avoided that one. My view is reviewer might not have increased the score.

M6wQ:

Authors claim ""While the gains are indeed more substantial in the fine-tuning setting, the from-scratch results outperform the baseline."" however those gains are quite small and Table-3 from scratch results do not outperform original. In Table-3 DAP required baseline or the box-regression to slightly improve on baseline result.
Authors agreed to many points that should be included in the main draft and would have satisfied that part of the review. However, on results part I believe reviewer might not have improved too much towards the positive side, might be 5.
"
mniJ


"Authors conceded on Song but presented argument defending their point on Henderson. They stepped away from the validation set request by presenting reasoning about 10% of pretraining length be their post-hoc training epochs. Results were not provided on other dataset.
Reviewer might have increased but not so much.

9XdG


generally happy with the responses provided and stated that it  will raise its recommendation accordingly.

**Reviewer Scores:**

9XdG would have increased the score, reviewer has reviewed previous versions of this drafts submitted to other conferences.
UHGD might not have. M6wQ and mniJ might have but not too much towards positive side, might be rank 5.

I went over the draft itself. Following are my points

Grammatical and readability needs to be improved (e.g. L72 2nd contribution). In Introduction when first time presenting DAP (Line 52), please include full name too with abbreviation. Some places Fig. is used and some places Figure, please use one of them. Table-1 numbers in green showing improvement should be shown in separate column. Figure 3 is not properly readable and should be increased in size, fonts should be adjusted, line should be thick, etc..

 Authors claim to improve on DETR in Introduction but fail to discuss effect on others non DETR methods.
Comparison is only with DETR based methods and only on CoCO (and one on LVIS).
Authors have listed why fine tuning works but not the training from scratch but have not given any proper mechanism to identify when to shift from original loss function to the DAP.
In-depth comparison and results are missing, where improvement is quite small but still its improvement (in rebuttal only 4 epochs got +1.3 vs +0.4 if DAP is not used), however, we do not know where this improvement comes from, long tail, small objects, difficult objects?


I share astonishment expressed by the 9XdG that paper going through multiple revisions still lack so much of writing improvements. Suggestions made by mniJ, M6wQ, 9XdG, portions of General Response, etc.. means substantial revision of the draft.

---

### Decision · Program_Chairs · 2026-01-26

Reject